# Attitudes and Behaviors towards Vaccination in Portuguese Nursing Students

**DOI:** 10.3390/vaccines11040847

**Published:** 2023-04-14

**Authors:** Cristina Maria Alves Marques-Vieira, Tiago Dias Domingues, Adriana Dutra Tholl, Rosane Gonçalves Nitschke, Francisco Javier Pérez-Rivas, María Julia Ajejas-Bazán, Maria Clara Roquette-Viana

**Affiliations:** 1Center for Interdisciplinary Research in Health, Institute of Health Sciences, Nursing School (Lisbon), Universidade Católica Portuguesa, Palma de Cima, Edificio 5, Piso 5, 1649-023 Lisbon, Portugal; 2Lisbon Center for Research, Innovation and Development in Nursing (CIDNUR), Higher School of Nursing, 1600-190 Lisbon, Portugal; 3Centro de Estatistica e Aplicações da Universidade de Lisboa (CEAUL), Faculdade de Ciências, Universidade de Lisboa, 1649-004 Lisbon, Portugal; tmdomingues@ciencias.ulisboa.pt; 4Department of Nursing, Universidade Federal de Santa Catarina, Florianópolis 88040-900, Brazil; adriana.dutra.tholl@ufsc.br (A.D.T.); rosanenitschke@ufsc.br (R.G.N.); 5Grupo de Investigación UCM “Salud Pública-Estilos de Vida, Metodología Enfermera y Cuidados en el En-torno Comunitario”, Departamento de Enfermería, Facultad de Enfermería, Fisioterapia y Podología, Universidad Complutense de Madrid, 28040 Madrid, Spain; frjperez@ucm.es (F.J.P.-R.); majejas@ucm.es (M.J.A.-B.); 6Red de Investigación en Cronicidad, Atención Primaria y Promoción de la Salud—RICAPPS—(RICORS), Instituto de Investigación Sanitaria Hospital 12 de Octubre (Imas12), 28041 Madrid, Spain; 7Academia Central de la Defensa, Escuela Militar de Sanidad, Ministerio de Defensa, 28040 Madrid, Spain

**Keywords:** attitudes, behavior, COVID-19, students, vaccination, nursing

## Abstract

Knowing the attitudes and behaviors of nursing students in relation to vaccination is important because they will soon be determinant for the health literacy of the population. Vaccination remains the most effective response in the fight against communicable diseases, including COVID-19 and influenza. The objective of this study is to analyze the attitudes and behaviors of Portuguese nursing students with regard to vaccination. A cross-sectional study was carried out, with data collection from nursing students at a university in Lisbon, Portugal. A sample of 216 nursing students was obtained, representing 67.1% of the students enrolled in this university. What stands out from the results of the questionnaire “Attitudes and Behaviors in Relation to Vaccination among Students of Health Sciences” is that for the majority of students the answers were positive; in addition, 84.7% had a completed vaccination schedule for COVID-19. Being a nursing student, being in the final years of the course and being a woman are the factors that most influence the positive attitude of the students. The results obtained are motivating, because these students will be the future health professionals most likely to integrate health promotion programs through vaccination.

## 1. Introduction

Vaccination is one of the most effective public health interventions in controlling communicable diseases [1,2,3]. From individual protection to herd immunity, the importance of vaccination goes far beyond the vaccinated person’s immunity against a specific disease [2,3,4]. In recent decades, vaccination programs have contributed to the reduction or even elimination of serious diseases (such as smallpox), to the long-term progress of communities (with lower mortality, better health conditions and socioeconomic benefits) and they are fundamental for the achievement of the sustainable development objectives of the World Health Organization (WHO) [1,2,3,5].

In addition to reducing mortality and morbidity rates, vaccination also helps to avoid clinical complications associated with various infections, which can often manifest themselves chronically throughout life [3]. It is crucial to remember that, in the absence of vaccines, these consequences can lead to further diseases, disabilities, and, in general, put health at risk, lowering the quality of life and affecting both the sick person and family caregivers [3]. It is up to health professionals, and in particular nurses, to motivate and take responsibility for carrying out their role as promoters of health education in the community [6,7]. Improving health in the community is one of the focuses of nursing interventions, as individual health is related to the health of the population as a whole [6,7]. In this context, it is important to add that the reinforcement of the individual’s empowerment and responsibility to contribute to the improvement of individual and community health is reinforced through health promotion by nurses, in a continuous dynamic of development that integrates the production and sharing of information/knowledge (health literacy), which is becoming increasingly necessary. Currently, a culture of proactivity, commitment, and self-control (training/active participation) is being promoted, for maximum individual and collective responsibility and autonomy (empowerment) [6,7,8].

Health promotion is the process of creating conditions for people to act on health determinants, either individually or collectively, in order to optimize health gains, contribute to the elimination of inequalities, and generate social capital [6,7]. Thus, promoting health means, not only at an individual level, but also at a community level, creating a health system that allows each person not only to prevent illness, but also to promote and protect their health [6,7].

Nursing students are encouraged to reflect on these issues during their training in order to gain the knowledge needed to build their skills as future nurses [9].

Considering the issue of health promotion to be of the utmost importance, mainly in the sequence of experiences experienced in the current context, and following studies carried out in Spain [10,11,12,13], the objective of the study was to analyze the attitudes and behaviors of nursing students in the face of vaccination, but within the particularities of the Portuguese context. Regarding particular objectives, the aim is to: determine influenza and COVID-19 vaccination coverage in a sample of nursing students; identify characteristics related to this vaccination coverage; and correlate attitudes and behaviors with regard to vaccination with age, gender, and year of study.

Thus, the goal is to instill in the student a critical and analytical awareness of the issues around them, enabling them to actively participate in social change and contribute to improvements in health, as well as to determine whether behaviors change over the course of the four years required to earn a nursing degree, during which time knowledge is consolidated and skills become consistent.

## 2. Materials and Methods

This study is quantitative, transversal, and inserted in the positivist paradigm. Its sample was 322 nursing students attending the Nursing Degree Course at the Lisbon School of Nursing in the Institute of Health Sciences of the Universidade Católica Portuguesa (Lisbon, Portugal), enrolled in the academic year 2021/2022, the year in which Portugal was beginning to ease measures related to the pandemic context (the declaration of the state of emergency in Portugal was on 18 March 2020).

All enrolled students, regardless of year, were invited to complete the Scale of “Attitudes and Behaviors in Relation to Vaccination among Students of Health Sciences” (ACVECS—according to its Spanish initials) [10]), previously translated by two translators, which resulted in an overlapping translation. The ACVECS scale, which aims to measure health students’ attitudes and behaviors with regard to vaccination, is made up of 24 items, the first 15 of which measure the “attitudes” dimension and the last 9, “behaviors” [10,11,12,13]. These 24 items were answered on a five-point Likert-type scale, ranging from 0 = strongly disagree to 4 = strongly agree. For items 1, 2, 7, 8, 15 and 23, the scores will need to be reversed before analysis, given the way the questions are worded [10,11,12,13]. Beliefs, behaviors and general attitude scores of ≥3 were considered positive (i.e., vaccination-favorable), ≤1 negative and =2 considered neutral or indifferent [12]. In the applied data collection instrument, preceding the scale itself, there was a space for the characterization of the sample (sex, age, year of the course, country of birth, if the respondent had been vaccinated against the flu and COVID-19). This model was a replica of the study carried out by researchers at the Universidad Complutense de Madrid (Madrid, Spain) [11]. The scale takes an average of 10 min to complete.

The researchers began by contacting the course coordinators, with subsequent contact with the teachers of the curricular units taking place in the week of 21 to 25 March 2022. After these authorizations, an awareness-raising event was carried out in the classroom, where the study objectives were presented. This was followed by all the information being sent by email. These procedures were intended to safeguard the partiality of the students’ responses. This email contained access to a link, which enabled them to answer the questionnaire online, using Google Forms. In this way, it was guaranteed that the answers were free and clarified.

Data were collected from 25 March to 31 May 2022. All the information collected was subsequently transferred to a database using Microsoft Office Excel 2016.

### Statistical Analysis

Quantitative variables were described as mean ± standard deviation (SD) (data normally distributed) or median within the respective interquartile range. The underlying normality of the data was assessed using a Kolmogorov–Smirnov test with a Lilliefors correction. Comparisons between two independent groups were performed using a t test for independent samples or a Mann–Whitney test; for more than two independent groups, comparisons were performed using the Analysis of Variance (ANOVA) using Tukey multiple pairwise comparisons. Regarding qualitative variables, these were described as n (%). A Fisher exact test was used to assess associations between qualitative variables.

In order to evaluate the impact of sex, age, course year, beliefs, behaviors, and general attitudes on vaccination against influenza and COVID-19, binary logistic regression was constructed considering the vaccination status (0—not vaccinated, 1—vaccinated) as the dependent variable and sex, age, course year, beliefs, behaviors and general attitude as covariates. A stepwise method was used to select the best binary logistic model. We computed the receiver operating characteristic (ROC) curve to differentiate between vaccinated cases and non-vaccinated cases. We then estimated the respective areas under these curves (AUC) and computed their 95% confidence interval. The Hosmer–Lemeshow test was also presented.

Statistical analysis was performed using the software R version 4.2.2. All the results with a *p*-value less than 0.05 were considered statistically significant.

## 3. Results

A sample of 216 nursing students who answered the ACVECS questionnaire was obtained, that is, 67.1% of a total population of 322. Of 215 students for whom information was obtained, 89.8% (*n* = 193) were female and 10.2% (*n* = 22) were male; one of the students did not answer regarding gender (female, male and other). Median age was 21 years (22.00–19.00) and 20.5 years (22.00–19.75) (Mann–Whitney test: *p* = 0.43) for females and males, respectively. Of the participants, 91.6% (*n* = 197) were born in Portugal and 8.4% (*n* = 18) were born abroad. Of these, 32.6% (*n* = 70) were in the first year of the nursing course, 26.9% (*n* = 58), 20.5% (*n* = 44) in the second or fourth year and 20.0% (*n* = 43) in the third year.

With regard to the scores attributed, most of the individuals surveyed adopted a positive attitude (i.e., score ≥ 3) (Table 1).

According to the distribution of the scores by the dimensions, there were no statistically significant differences between men and women, which means that beliefs and behaviors are homogeneous between individuals. However, there are variations in the scores according to the year of the course they are attending with regard to beliefs and general attitude (Table 2). In the case of beliefs, the differences were observed between the first and third years (adjusted *p*-value from Tukey multiple pairwise comparisons: *p* = 0.02) and the second and third years (adjusted *p*-value from Tukey multiple pairwise comparisons: *p* < 0.01). Regarding general attitudes, differences were observed between the second and third years (adjusted *p*-value from Tukey multiple pairwise comparisons: *p* = 0.006).

Regarding vaccination, it was found that 85.2% (*n* = 184) did not get vaccinated against influenza. No differences were observed between males and females, respectively (Fisher exact test: 86.4% vs. 85.5%; *p* > 0.1). And in relation to vaccination against COVID-19, 84.7% of the students had a complete vaccination schedule. Table 3 and Table 4 are intended to summarize the information from the univariable and multivariable binary logistic (best model) regression results considering flu vaccination and COVID-19 as dependent variables, respectively.

Considering the impact of the variables—gender, age, course year, beliefs, behaviors, and general attitude—on influenza vaccination (Table 3), it was found that a positive attitude toward vaccination increases with age (OR = 1.11 (1.02–1.21); *p* = 0.01). This trend has an impact on the year of the course the students are attending; therefore, fourth year students are more sensitive to the vaccination issue (OR = 5.35 (1.69–20.28); *p* = 0.006). A positive attitude toward vaccination effectively makes it more likely that the individual will get vaccinated (OR = 10.54 (3.36–39.13); *p* < 0.01). The model discriminates the data effectively (Hosmer–Lemeshow statistic: 13.74; *p* = 0.09; AUC = 0.80, CI 95%: 0.72–0.88). Regarding the COVID-19 vaccination (Table 4), it was found that the vaccination trend was seen in younger individuals (OR = 0.88 (0.81–0.85); *p* < 0.01), and this vaccination trend is observed in the third (OR = 5.09 (1.32–24.12); *p* = 0.03) and fourth years (OR = 13.28 (3.13–85.67); *p* < 0.01) of the course. A positive attitude toward the vaccination makes them get vaccinated more (OR = 10.79 (3.64–38.00); *p* < 0.01). Being female makes it four times more likely that they will get vaccinated (OR = 4.01 (0.96–16.33); *p* = 0.05). It was also observed that individuals who do not have a confirmed diagnosis of COVID-19 are more likely to be vaccinated (OR = 0.13 (0.03–0.38); *p* < 0.01), which reveals a greater awareness of vaccination. The model discriminates the data effectively (Hosmer–Lemeshow statistic: 11.11; *p* = 0.196; AUC = 0.86, CI 95%: 0.79–0.93).

## 4. Discussion

In this study, the attitudes and behaviors of nursing students in Portugal, more specifically in Lisbon, regarding vaccination and self-reported flu and COVID-19 vaccine coverage, were investigated. It should be noted that in Portugal it is not mandatory to take the flu vaccine; it is only recommended for more vulnerable groups, such as the elderly, people with certain associated pathologies (respiratory diseases, etc.), and health professionals, among others. In the pandemic context, there was no specific indication for nursing students, either regarding the flu vaccine, or in relation to the vaccination against COVID-19, not constituting the priority groups, in a first phase of the vaccination. When the study was carried out, all these issues had already been dealt with, and students, in order to carry out an internship, had to prove that they were vaccinated against COVID-19.

The “Attitudes and Behaviors Regarding Vaccination Among Health Sciences Students” Scale (ACVECS), validated by Fernández-Prada et al. [10], showed adequate scores (score ≥ 3) for positive attitudes.

Regarding the scores attributed to the questionnaire items for attitudes and behaviors related to vaccination, it was observed that, for 21 (87.5%) of the 24 items, the students’ responses were positive and greater than 70%. A study developed in Madrid [11] found a similar result, with 20 items above 70%. In the United States [14], high scores on attitudes and knowledge were achieved by 74.3% of medical students, 62.7% of pharmacy students, 57.1% of doctoral nursing students, and 24.7% of undergraduate nursing students. The authors conclude that negative attitudes, lack of knowledge and general discomfort exist in all programs, but in relation to vaccination, they occur especially in nursing [14], as opposed to the results obtained in Portugal and Spain [11,12,13]. In this sense, it is important to highlight that some factors can interfere with the effectiveness of the health planning process, namely behavior. In the Portuguese National Health Plan 2021–2030, it is stated that a “new way of thinking and behaving” is needed for health planners, adequate for a context of uncertainty and high complexity ([8], p. 33).

In a cross-sectional study carried out in Poland [15], with several health professionals (*n* = 1137), it was identified that knowledge of vaccination and vaccination coverage among students on four courses (Medicine, Nursing, Pharmacy and Public Health) was low, and the main factors associated with a negative attitude toward vaccination were feelings of good health and the age factor [15]. On the other hand, in other European studies [12,15], the factors that most influenced the positive attitude toward vaccination were: being a nursing student, being female [16] and attending the last years of the course [11,17]. The authors conclude that being a nursing student can be a positive aspect because they are more involved, have greater responsibilities in vaccination programs and receive more information and clinical practice in vaccination than students on other courses. An important aspect to be considered in this study is the fact that a high percentage of nursing students responded by saying that the obligation to vaccinate against flu was not ethical. Negative (≤1) and null (=2) scores were 25.1 and 23.7%, respectively. Regarding flu vaccination status, we also identified that the chance is one time higher among older and more advanced individuals in relation to youngest [OR = 1.11 − (CI 95%-1.02–1.21)], as well for the individuals in advanced nursing course [OR = 5.35 (1.69–20.28)]. These findings corroborate several studies [11,12,18,19].

It can be observed in this sample that 84.7% (*n* = 183) had the vaccination schedule for COVID-19 complete. In an Italian study of 934 medical and nursing students, it was observed that 86.1% were willing to be vaccinated against COVID-19 [20]. In the United States, it was observed that 92% of nursing students and professors had a positive attitude toward vaccination [21]. In Brazil, an online survey of adults identified that vaccine hesitancy occurred in only 10.5% of respondents and that only 1.3% had no intention of getting vaccinated [19].

A pioneering study [22] in seven European countries, with the aim of exploring nursing students’ intentions to get vaccinated against SARS-CoV-2 infection, identified that the main reasons for wanting to be vaccinated were being male, not having worked in health facilities during the pandemic, having been vaccinated against flu in 2019 and 2020, trust in doctors, governments and specialists, a higher level of knowledge and fear about COVID-19 [23]. A study of medical students showed that 77.0% of the population had a positive attitude toward the vaccine after obtaining confirmation of its efficacy [24].

In our study, we found that 84.7% had a complete vaccination schedule. According to data from Our World in data [25], 86.6% of the population of Portugal received a vaccination for COVID-19, that is, the two-dose schedule for most vaccines, with 95.2% receiving at least one dose of the vaccine. According to data collected globally, 69.7% of people worldwide have received a dose of the COVID-19 vaccine.

Despite international agencies [26] recommending influenza vaccination for health science students, only 14.8% of respondents had been vaccinated. This coverage is significantly lower than that reported in other international studies of health science students [13,27,28,29,30]. There is a need to discuss what strategies can be implemented to increase this coverage among students, especially among those on training placements in health care settings.

Immunization has resulted in a dramatic decline in morbidity and mortality from infectious diseases [1]. However, there is an increasing rate of vaccine hesitancy. Acceptance of the vaccine among the general public was very low, with countries reporting less than 50% of expected coverage in target populations and worldwide [19]. Furthermore, vaccine hesitancy is not only present in the general population, but also among health professionals, and has been shown to influence their behavior and vaccine-related decisions [31].

It is important to emphasize that health professionals, in addition to administering vaccines, play a crucial role in promoting health literacy in relation to immunizations [11], which leads to the empowerment of the population. Interprofessional education, that is, those activities that involve two or more health professionals, can also improve collaboration and the quality of care provided, and be predictive of their attitudes, beliefs, and behaviors as health professionals [13].

Finally, the present work has some limitations. It is a voluntary study using a self-administered questionnaire and the results are based on the responses of the participants. Although those who responded represented 67.1% of all students, no information was collected on those who did not participate. It is possible that their characteristics were different from those of the participants and that a non-response bias may exist. The comparison of responses regarding the attitudes and behaviors of nursing students over the four years should be carefully evaluated, because these are students with different specificities, and at different stages of learning, which is why it is recommended that a longitudinal study be carried out. The outcomes are based from one university only, in one city (Lisbon).

The results must be interpreted with caution, because the sample is not representative of Portuguese nursing students, or of nursing students from other institutions. It is recommended that studies be carried out in other universities in Portugal.

## 5. Conclusions

The COVID-19 vaccine responded to a pandemic experienced worldwide. This experience led to greater awareness and adherence of the population to the importance of vaccination programs, in view of the prevention and control of certain diseases. We are once again experiencing the emergence of new strains worldwide, where possible new vaccination proposals will emerge. Nursing students, as future nurses, will have a leading role in education on the health of populations, more specifically in health literacy.

With this study, it was concluded that, in this sample of nursing students, beliefs, behavior, and general attitude toward vaccination, were positive and that vaccination coverage against COVID-19 was high, with no significant differences in relation to sex. These results are important because these students will be the future health professionals integrating health promotion programs through vaccination and supporting decision-making regarding health policies. The findings of this study will be useful in assisting teachers in the teaching–learning process, meeting the specific needs of students and guiding them in identifying the problems and difficulties that arise, determining the aspects to be observed and on which to reflect and establish the appropriate strategies, defining action plans to be followed, and finally, in the creation of professional knowledge. This study aroused a greater awareness of the impact of the problems of the community in which the academy operates, making us effective instruments in a logic of social responsibility, with the achievement of health gains.

In the future, the intention is to replicate this study in other countries and compare the results obtained.

## Figures and Tables

**Table 1 vaccines-11-00847-t001:** ACVECS questionnaire: frequencies and percentages of students showing negative, neutral and positive scores.

Questionnaire Items	Negative n (%)	Neutral n (%)	Positive n (%)
1- I have doubts about the effectiveness of vaccines	11 (5.1)	15 (6.9)	190 (88.0)
2- I would rather have influenza than be vaccinated against it	28 (13.0)	30 (13.9)	158 (73.1)
3- I am convinced that marketed vaccines are safe	6 (2.8)	23 (10.6)	187 (86.6)
4- I am interested in learning more about vaccination	5 (2.3)	17 (7.9)	194 (89.8)
5- I believe it is important to check my vaccination status before travelling to a tropical country such as Mexico or Thailand	1 (0.46)	1 (0.46)	214 (99.08)
6- National and international vaccine campaigns are cost-effective	8 (3.7)	34 (15.8)	173 (80.5)
7- It is not worth being vaccinated against a disease for which effective treatment exists	10 (4.6)	15 (7.0)	191 (88.4)
8- Vaccinating the adult population is not important	2 (0.9)	0 (0.0)	214 (99.1)
9- Health science students are ethically obliged to be vaccinated against influenza	54 (25.1)	51 (23.7)	110 (51.2)
10- Being vaccinated myself has a positive influence on the behavior of my patients	9 (4.1)	36 (16.7)	171 (79.2)
11- Students should be vaccinated to reduce the transmission of infectious diseases in hospital	4 (1.8)	19 (8.8)	193 (89.4)
12- I should review my vaccination status before starting clinical training	9 (4.2)	27 (12.5)	180 (83.3)
13- I should be vaccinated against influenza every year, even if it means missing hours of practical training	37 (17.2)	46 (21.4)	132 (61.4)
14- I would be vaccinated irrespective of what my peers might do	11 (5.1)	18 (8.3)	187 (86.6)
15- If I am in good health, there is no need to be vaccinated	9 (4.2)	15 (6.9)	192 (88.9)
16- I would recommend my patients adhere to the established vaccination calendar	2 (0.9)	1 (0.5)	212 (98.6)
17- I would inform my patients of the effectiveness, indications, and side effects of each vaccine	2 (0.9)	1 (0.5)	210 (98.6)
18- I would travel to a tropical country only after consulting an international vaccination center	6 (2.8)	10 (4.6)	199 (92.6)
19- I would be vaccinated against HIV when a vaccine becomes available and when it is shown to be acceptably safe and effective	4 (1.9)	18 (8.3)	194 (89.8)
20- If being vaccinated against influenza were readily accessible to me, I would be vaccinated every year	23 (10.7)	25 (11.6)	167 (77.7)
21- I would be vaccinated against anything my doctor recommends, even if I have to pay for it	24 (11.2)	44 (20.5)	147 (68.3)
22- When I begin work at a hospital, I will make sure I am vaccinated against everything preventable	3 (1.4)	10 (4.7)	202 (93.9)
23- I would only be vaccinated in exceptional circumstances (epidemics, health alerts, etc.)	11 (5.1)	14 (6.5)	191 (88.4)
24- I will be vaccinated against influenza every year I have clinical training	29 (13.4)	33 (15.3)	154 (71.3)

**Table 2 vaccines-11-00847-t002:** Distribution of scores for ACVECS questionnaire dimensions with respect to sex and nursing course year.

Variables	Categories	Beliefs Mean ± SD	*p*	Behavior Mean ± SD	*p*	General Attitude Mean ± SD	*p*
Sex	Male	3.25 ± 0.42	0.94	3.34 ± 0.66	0.83	3.28 ± 0.46	0.96
	Female	3.24 ± 0.46		3.37 ± 0.54		3.29 ± 0.46	
Nursing course year	1st	3.28 ± 0.42	<0.01	3.34 ± 0.55	0.55	3.30 ± 0.44	0.01
	2nd	3.44 ± 0.39		3.47 ± 0.45		3.45 ± 0.39	
	3rd	3.04 ± 0.45		3.31 ± 0.50		3.14 ± 0.45	
	4th	3.22 ± 0.44		3.36 ± 0.65		3.28 ± 0.49	

**Table 3 vaccines-11-00847-t003:** Univariable and multivariable binary logistic (best model) regression results considering influenza vaccination status as the dependent variable.

Variables	β	OR (CI 95%)	*p*	β	OR (CI 95%)	*p*
Sex						
*Male vs. female*	0.072	1.07 (0.34–4.78)	0.91			
Age (years)	0.089	1.09 (1.02–1.17)	<0.01	0.10	1.11 (1.02–1.21)	0.01
Course year						
*2nd vs. 1st*	0.72	2.05 (0.58–7.56)	0.26	0.75	2.11 (0.53–9.51)	0.29
*3rd vs. 1st*	0.75	2.11 (0.59–7.77)	0.24	1.27	3.58 (0.83–16.49)	0.09
*4th vs. 1st*	1.51	4.53 (1.63–14.78)	<0.01	1.68	5.35 (1.69–20.28)	<0.01
Beliefs	1.91	6.76 (2.49–20.69)	<0.01	2.36	10.54 (3.36–39.13)	<0.01
Behavior	1.24	3.45 (1.46–9.35)	<0.01			
General attitude	1.87	6.46 (2.32–20.54)	<0.01			

Legend: β—Coefficient.

**Table 4 vaccines-11-00847-t004:** Univariable and multivariable binary logistic (best model) regression results considering COVID-19 vaccination status as the dependent variable.

Variables	β	OR (CI 95%)	*p*	β	OR (CI 95%)	*p*
Sex						
*Male vs. female*	1.09	2.99 (1.06–7.85)	0.03	1.39	4.01 (0.96–16.33)	0.05
Age (years)	−0.07	0.93 (0.87–0.99)	0.04	−0.13	0.88 (0.81–0.95)	<0.01
Course year						
*2nd vs. 1st*	0.29	1.33 (0.53–3.59)	0.55	−0.43	0.65 (0.19–2.11)	0.47
*3rd vs. 1st*	1.06	2.89 (0.97–10.68)	0.08	1.63	5.09 (1.32–24.12)	0.03
*4th vs. 1st*	1.39	4.00 (1.36–14.66)	0.02	2.59	13.28 (3.13–85.67)	<0.01
Beliefs	1.54	4.68 (2.03–11.41)	<0.01	2.38	10.79 (3.64–38.00)	<0.01
Behavior	1.04	2.84 (1.51–5.62)	<0.01			
General attitude	1.51	4.51 (2.02–10.78)	<0.01			
COVID-19 diagnosis	−1.54	0.22 (0.08–0.51)	<0.01	−2.07	0.13 (0.03–0.38)	<0.01

Legend: β—Coefficient.

## Data Availability

The datasets generated and/or analysed during the current study are not publicly available in order to preserve patients’ confidentiality, but are available from the corresponding author on reasonable request.

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
