# Peer review of "Attitudes and Behaviors towards Vaccination in Portuguese Nursing Students"

_vaccines, 2023, doi:10.3390/vaccines11040847_

Round 1
Reviewer 1 Report
REVIEWER'S REPORT
Manucsript title: Attitudes and Behaviors towards Vaccination in Portuguese Nursing Students (Authors: Cristina Maria Alves Marques-Vieira, Tiago Dias Domingues, Adriana Dutra Tholl, Rosane Gonçalves Nitschke, Francisco Javier Pérez-Rivas, María Julia Ajejas-Bazán and Maria Clara Roquette-Viena).
The purpose of this study was to examine the vaccination attitudes and behaviors of Portuguese nursing students. Data were collected from nursing students at a university in Lisbon, as part of a cross-sectional study. This manuscript, in my opinion, can be accepted without a doubt with small text alterations, that could increase the paper's quality.
In Introduction (page 2), the sentence (in lines 51-55) should be paraphrased as " It is crucial to remember that, in the absence of vaccines, these consequences can lead to additional diseases, disabilities, and, in general, put health at risk, lowering quality of life and hurting both the sick person and family caregivers." The end of sentence (in lines 62-63) should be written "..., which is becoming increasingly necessary." The sentence in lines 66-68, must be written as " Health promotion is the process of creating conditions for people to act on health determinants, either individually or collectively, in order to optimize health gains, contribute to the elimination of inequities, and generate social capital." The sentence in lines 72-73, might be witten as " Nursing students are encouraged to reflect on these issues during their training in order to gain the knowledge needed to build their skills as future nurses." The sentence (in lines " Regarding particular objectives, it is aimed to: determine influenza and COVID-19 vaccination coverage in a sample of nursing students; identify characteristics related with this same vaccination coverage; and correlate attitudes and behaviors toward vaccination with age, gender, and year of study." The sentence (in lines 83-87) should be written " Thus, the goal is to instill in the student a critical and analytical awareness of the issues around them, enabling them to actively participate in social change and contribute to improvements in health, as well as to determine whether behaviors change over the course of the four years required to earn a nursing degree, during which time knowledge is consolidated and skills become consistent."
In Discussion (page 7), the sentence (in lines 219-220) needs to be written as "According to data collected globally, 69.7% of people worldwide have received a dose of the COVID-19 vaccine".
In Conclusion (page 7), the final sentence in lines 254-257 should be written as "The findings of this study will be useful in assisting teachers in the teaching-learning process, meeting the specific needs of students and guiding them in identifying the problems and difficulties that arise, determining the aspects to be observed and on which to reflect and establish the appropriate strategies, defining action plans to be followed, and finally, in the creation of professional knowledge."
This work would be considerably more appealing if the key statistical results were also displayed in the form of graphs (eg histograms, etc.).
Author Response
Thank you for the rigorous review, which allows us to improve our article.
In order to facilitate the perception of the changes incorporated in the article, we carried out a cover letter to explain, point by point, the details of the revisions to the manuscript and your responses to the referees' comments.
Best wishes,
Cristina Marques Vieira

Reviewer 2 Report
The authors should answer a number of questions:
1. why do so high a % of students answer positively to the questions below in Table 1?
a. I have doubts about the effectiveness of vaccines 88% positive
b. I would rather have influenza than be vaccinated against it 73% positive
c. It is not worth being vaccinated against a disease for which effective treatment exists 88% positive
2. The answers and results presented in Table 1 do not correspond with the authors' conclusions.
3. The authors should change the first sentence in the discussion - the analysis concerns only one University in one city
4. What does the term Portuguese context mean?
5. The authors did not provide clear data on anti-influenza vaccination among the study group and in Portugal in general. Are the students required to have a flu vaccination?
6. The authors should provide data on whether students were required to be vaccinated against COVID as in other countries? Such an obligation affects the vaccination results of the study group.
7. The authors should take into account the fact that as many as 85% of students were not vaccinated against flu, which reflects their opinions on vaccination in general. No data on flu incidence in the study group is available.
- 8. The cited paper No. 15 did not apply only to Poland
9. The authors do not draw the right conclusions. The results indicate that about 80% of students have no intention to vaccinate, and only mandatory vaccination against COVID gave different results.
The paper does not touch on new topics, it only indicates low awareness of the need for immunization even in Portugal.
Author Response

(The authors gave the same response as above.)

Reviewer 3 Report
This is an amazing study. I have several minor comments.
1. Please specify what type of vaccine in the title and abstract.
2. I'd bring the context of COVID-19 and influenza earlier in Introduction.
3. Data analysis should be discussed as part of Methods. Describe and discuss the choice of your methodology - logit models.
4. I'd add more concrete implications on professional education based on study findings. What do these findings mean if you are implementing new educational curriculum?
5. Would you discuss selection bias as part of limitation? How can we address this issue?
Author Response

(The authors gave the same response as above.)

Round 2
Reviewer 2 Report
Dear Cristina Marques Vieira,
After being supplemented with critical comments in the discussion, the work can be published.